# Clinical and social determinants of health features of SARS-CoV-2 infection among Black and Caribbean Hispanic patients with heart failure: The SCAN-MP Study

Jonathan B. Edmiston[1], Elizabeth G. Cohn[2], Sergio L. Teruya[3], Natalia Sabogal[1], Daniel Massillon[1], Varsha Muralidhar[1], Carlos Rodriguez[3], Stephen Helmke[3], Denise Fine[1], Morgan Winburn[1], Codruta Chiuzan[4], Eldad A. Hod[5], Farbod Raiszadeh[6], Damien Kurian[6], Mathew S. Maurer[3], Frederick L. Ruberg[1]*

1 Section of Cardiovascular Medicine, Department of Medicine, Boston Medical Center, Boston University School of Medicine, Boston, Massachusetts, United States of America, 2 Hunter College, City University of New York, New York, New York, United States of America, 3 Seymour, Paul, and Gloria Milstein Division of Cardiology, Department of Medicine, Columbia University Irving Medical Center, New York, New York, United States of America, 4 Feinstein Institute for Medical Research, Northwell Health, New York, New York, United States of America, 5 Department of Pathology and Cell Biology, Columbia University Irving Medical Center, New York-Presbyterian Hospital, New York, New York, United States of America, 6 Division of Cardiology, Harlem Hospital Center, New York City Health and Hospital Corporation, New York, New York, United States of America

* frruberg@bu.edu

**Data Availability Statement:** All relevant data are within the paper and its Supporting information files.

## Abstract

Patients with heart failure (HF) often have multiple chronic conditions and are at increased risk for severe disease and mortality when infected by SARS-CoV-2, the virus that causes COVID-19. Furthermore, disparities in outcomes with COVID-19 have been associated with both racial/ethnic identity but also social determinants of health. Among older, urban-dwelling, minority patients with HF, we sought to characterize medical and non-medical factors associated with SARS-CoV-2 infection. Patients with HF living in Boston and New York City over 60 years of age participating in the Screening for Cardiac Amyloidosis with Nuclear Imaging (SCAN-MP) study between 12/1/2019 and 10/15/2021 (n = 180) were tested for nucleocapsid antibodies to SARS-CoV-2 and queried for symptomatic infection with PCR verification. Baseline testing included the Kansas City Cardiomyopathy Questionnaire (KCCQ), assessment of health literacy, biochemical, functional capacity, echocardiography, and a novel survey tool that determined living conditions, perceived risk of infection, and attitudes towards COVID-19 mitigation. The association of infection with prevalent socio-economic conditions was assessed by the area deprivation index (ADI). There were 50 overall cases of SARS-CoV-2 infection (28%) including 40 demonstrating antibodies to SARS-CoV-2 (indicative of prior infection) and 10 positive PCR tests. There was no overlap between these groups. The first documented case from New York City indicated infection prior to January 17, 2020. Among active smokers, none tested positive for prior SARS-CoV-2 infection (0 (0%) vs. 20 (15%), p = 0.004) vs. non-smokers. Cases were more likely to be taking ACE-inhibitors/ARBs compared to non-cases (78% vs 62%, p = 0.04). Over a mean

**Funding:** This study was funded by NIH/NHLBI R01HI139671 and R01HL139671S1 to FLR and MSM. The funders had no role in study design, data collection and analysis, decision to publish, or preparation of the manuscript.

**Competing interests:** The authors have declared that no competing interests exist.

follow-up of 9.6 months, there were 6 total deaths (3.3%) all unrelated to COVID-19. Death and hospitalizations (n = 84) were not associated with incident (PCR tested) or prior (antibody) SARS-CoV-2 infection. There was no difference in age, co-morbidities, living conditions, attitudes toward mitigation, health literacy, or ADI between those with and without infection. SARS-CoV-2 infection was common among older, minority patients with HF living in New York City and Boston, with evidence of infection documented in early January 2020. Health literacy and ADI were not associated with infection, and there was no increased mortality or hospitalizations among those infected with SARS-CoV-2.

## Introduction

The outbreak of SARS-CoV-2, the virus that causes COVID-19, quickly spread from Wuhan, China to the United States. The first cluster in Wuhan was identified December 12, 2019 and the first officially recognized U.S. case was identified January 18, 2020 in Washington State. It was quickly observed that racial and ethnic minorities, including African Americans and Latinos, as well as those with cardiovascular comorbidities including hypertension, coronary artery disease, and heart failure, were disproportionately affected by COVID-19 morbidity and mortality [1, 2]. Whether these observed disparities in outcome were owing to a higher proportion of co-morbid conditions among minority patients or systemic structural barriers to care was unclear.

Prior studies evaluating COVID-19 in heart failure populations have almost entirely been reported among hospitalized patients, which does not include those with asymptomatic SARS-CoV-2 infection or mild-COVID [3–6]. The use of hospitalized cases and controls subjects many studies to selection bias, along with lack of adequate adjustment for confounding, particularly when propensity score matching is not utilized [7–9]. A South Korean study did address these issues by using a nationwide COVID-19 dataset and using propensity score matching, but likely missed many asymptomatic and mild-to-moderate symptom patients [10]. Throughout the pandemic, and especially during the first wave, there was a substantial proportion of the population who did not present for testing or their health care facilities did not have access to COVID-19 tests. Thus, data informing pre-infection characteristics are lacking in the non-hospitalized, community dwelling heart failure population.

The Screening of Cardiac Amyloidosis with Nuclear Imaging (SCAN-MP) study is a National Institutes of Health funded prospective cohort study of community dwelling, older, self-identified Black or Hispanic patients with heart failure. The study is primarily designed to explore hypotheses related to cardiac amyloidosis (an under-diagnosed cause of HF in this population) prevalence and course. Most SCAN-MP participants do not have cardiac amyloidosis and thus comprise an important cohort that can be leveraged to study HF in an entirely minority population.

Using serum nucleocapsid antibody testing to identify prior infection (along the full spectrum illness severity) and electronic health record review of diagnostic PCR COVID-19 testing, we sought to ascertain the proportion of participants enrolled in SCAN-MP infected with SARS-CoV-2 and how they may have differed from those who avoided infection. SCAN-MP participants are all urban-dwelling self-identified Black and Hispanics of Caribbean descent over 60 years of age who had predominantly mild to moderate heart failure often in the setting of a preserved ejection fraction, and residing in epicenters of COVID's first wave of infection–New York City and Boston. Patients were being enrolled through the early stages of

the pandemic, prior to recognized community spread. Our hypothesis was that infection rates would be the highest in socio-economically deprived parts of these urban centers, among those with low health literacy, and among those who did not appreciate the risk of infection and/or were not able to comply with CDC recommendations such as social distancing and masking. While outcomes (death, hospitalizations) were of interest, it was understood that the presence of antibodies to SARS-CoV-2 wouldn't necessarily be associated with an increase in death or hospitalizations, as it was a marker of past infection.

## Materials and methods

### Study population

The Screening for Cardiac Amyloidosis Using Nuclear Imaging for Minority Populations (SCAN-MP) study is a National Institutes of Health/National Heart, Lung, and Blood Institute funded, prospective cohort study that is enrolling Black or Caribbean Hispanic participants over the age of 60 years with heart failure. Heart failure (HF) was defined using either the modified criteria by Rich et al. or the National Health and Nutrition Examination Survey (NHANES) criteria with a score greater than or equal to 3 [11, 12]. Additional inclusion criteria are echocardiographic evidence of left ventricular septal or inferolateral wall thickness $\geq$ 12 mm and left ventricular ejection fraction greater than 30%. Exclusion criteria include evidence of primary (AL) or secondary (AA) amyloidosis, prior heart or liver transplantation, life expectancy < 1 year, heart failure due to left sided valvular disease or ischemic cardiomyopathy, end-stage heart failure (expected VAD or inotropes), stage V kidney disease (eGFR< 15/ mL/min/1.73m2), serious functional or cognitive impairment (e.g., stroke, injury, dementia, behavioral disorders) precluding study participation, and/or enrollment in a clinical trial not approved for co-enrollment. The primary aim of SCAN-MP is to assess the overall prevalence of transthyretin cardiac amyloidosis (ATTR-CA) in Black and Hispanic Caribbean patients with HF. SCAN-MP began enrollment in May 2019 and paused enrollment for 4 months starting in March 2020 owing to the first COVID-19 pandemic surge. The current study represents a sub-study of SCAN-MP funded by a specific award from National Institutes of Health/ National Heart, Lung, and Blood Institute (awards R01 HL139671 and HL139671S1). The study was approved by a single-institutional review board (IRB) mechanism with the Western Institutional Review Board (study ID 1252530, Protocol #20183425) as the reviewing organization and subsequent approval by the respective IRB's at Boston University Medical Campus, Columbia University Medical Center, and Harlem Hospital. All subjects provided written informed consent.

This study consisted of two recruitment cohorts followed prospectively. (Fig 1) describes the overall subject recruitment process for this study. First, we retrospectively identified subjects enrolled between December 1, 2019 and October 13, 2020 (the timing of study initiation) to invite participation. There had been n = 87 potentially eligible for participation, but four had died. Next, n = 83 were contacted (or contact was attempted) and n = 55 agreed to participate. Second, we prospectively invited all SCAN-MP subjects from October 14, 2020 to October 14, 2021 to participate. Of 150 eligible to enroll, 126 agreed to participate in this sub-study. In total, of 237 eligible participants, n = 181 agreed to participate. Of these, n = 178 of those participants had baseline nucleocapsid antibodies to SARS-CoV-2 measured from stored serum samples through an ELISA method constructed at Columbia University Medical Center. Nucleocapsid antibodies indicate prior infection and not prior vaccination (which elicits antibodies to the spike protein). Additionally, 135 participants were queried between their baseline, six-month, and twelve-month follow-up visits if they had tested positive for COVID-19, and if yes, the results of their SARS-CoV-2 polymerase chain reaction (PCR) test was

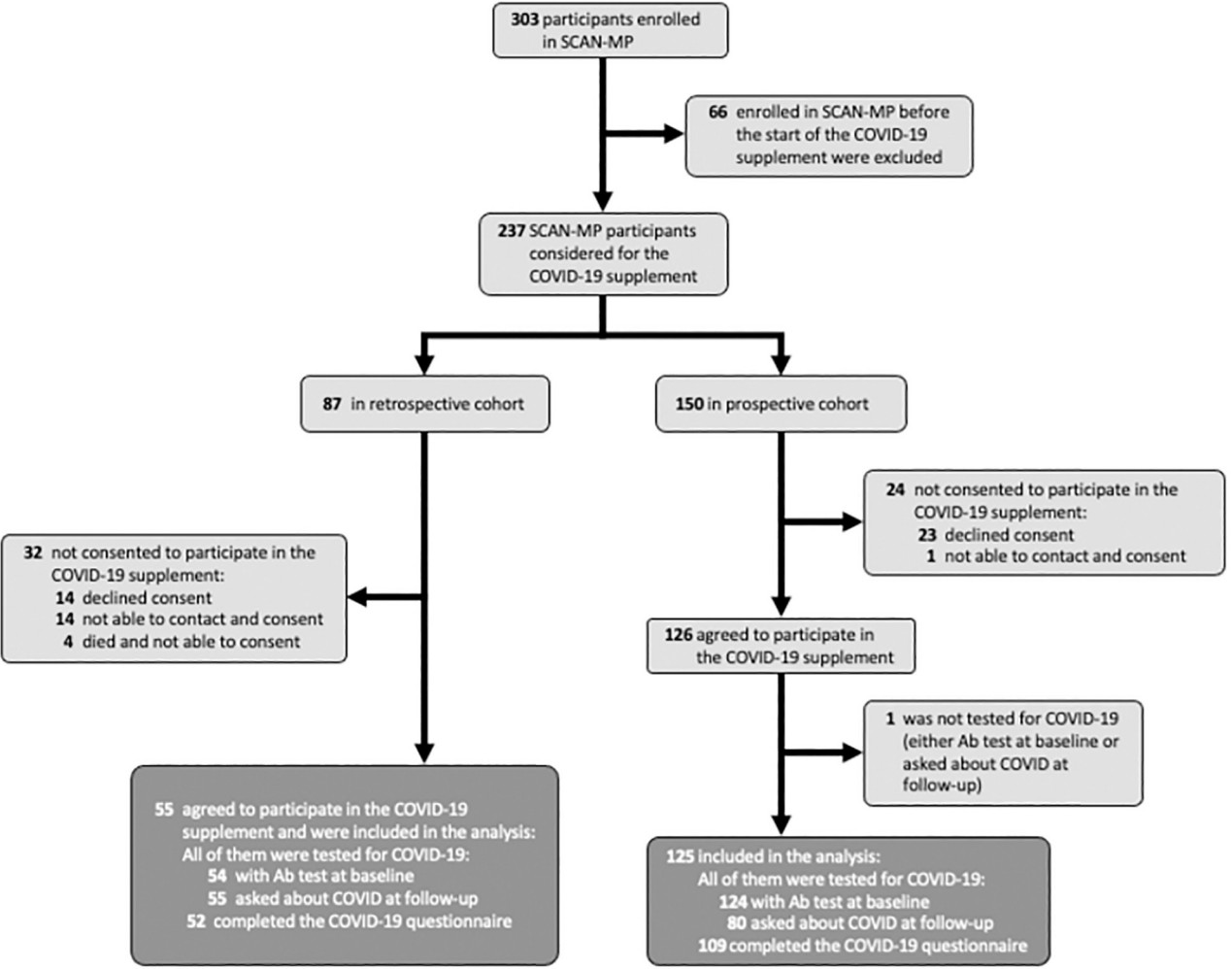

**Fig 1. Population flow chart.** Chart illustrates enrollment process, exclusions, and final study population. Note participants were enrolled retrospectively (after SCAN-MP enrollment but before commencement of the sub-study) as well as prospectively.

obtained. One of the subjects had neither a baseline antibody test nor was asked in the intervening visits about COVID, and therefore was excluded. At the baseline visit, 161 subjects also completed a *de novo* generated comprehensive questionnaire (S1 File) to assess the domains of knowledge of COVID-19 transmission and symptoms, perceived risk and attitudes towards the COVID-19 pandemic, and their sources of information about the COVID-19. Patients were also queried whether they had been hospitalized at any time in the six-months prior to their baseline visit. For those subjects who died during the study period, a call was made to family members to determine the cause of death.

The baseline visit consisted of a basic physical exam, six-minute walk test (6MWT), short physical performance battery (SPPB), Kansas City Cardiomyopathy Questionnaire (KCCQ), Health Literacy Questionnaire [13], Trust in Physician Scale Questionnaire [14], Perceived Discrimination Questionnaire [15], and English Proficiency questionnaires. Relevant laboratory testing included were complete metabolic panel, brain natriuretic peptide (BNP), N-terminal-pro-BNP (NT-proBNP), high sensitivity troponin-I (HS-TnI) and -troponin-T (HS-TnT) with cardiac testing including an electrocardiogram (ECG) and transthoracic echocardiogram.

Area Deprivation Index (ADI) was obtained from already computed ADI scores from Neighborhood Atlas [16] as previously described [17, 18]. The ADI was collected for census block group (BG) to include all zip codes present in the cohort. A census block is the smallest geographic unit, and BGs have unique numerical identity within the census tract. They never cross state, county, or census tract boundaries. The BGs are assigned a ranking based upon the ADI. The ADI national composite scale (1–100) where 1–10 is least disadvantaged and 90–100 is most disadvantaged. The BGs with higher ADI ranking are most disadvantaged. ADI normalized by decile by state for Massachusetts and New York were compared to national composite scale and found to have a correlation coefficient of 0.93. Given the normal distribution and high coefficient of correlation with the national composite scale, we selected the state decile ADI for analysis.

## Statistical analysis

All baseline exam, labs, imaging, and questionnaire data was analyzed with Stata 16.1 (StataCorp LLC, College Station, TX, USA). Means for continuous variables were compared using independent group *t*-tests when data were normally distributed; otherwise, Mann-Whitney tests were performed [19]. The normality of distributions was assessed using the Shapiro-Wilk test. Proportions for categorical variables were compared using the chi-squared test or Fisher's exact test, as appropriate. The hazard ratio for death was calculated using a Cox proportional hazards model with survival during follow-up assessed using Kaplan-Meier analysis [20]. All data were entered into RedCap and then exported for analysis using STATA version 16.1 1 (StataCorp LLC, College Station, TX, USA).

## Results and discussion

Of the 180 participants enrolled in this sub-study of SCAN-MP (Fig 1), there were 50 cases of SARS-CoV-2 infections (28% positive rate) with 40 having antibodies to SARS-CoV-2 at their baseline visit, and the remaining 10 having a positive PCR test between their baseline visit and follow-up. There were 108 participants (60% of total enrolled) from the NYC site, with the remainder recruited in Boston. There were 40 cases in NYC (37% positive rate) and 10 in Boston (14% positive rate) (p = 0.003). The first positive case in our cohort was from a blood sample collected on January 17, 2020 (indicating infection prior to this date), in NYC, with 3 additional cases in NYC later in January of 9 total, rendering a 44% positive rate for patients enrolled in NYC in January, 2020. The first Boston case was March 4, 2020. The case rate had no clear trend in Boston or NYC (Fig 2). The main SCAN-MP study was closed from mid-March to June of 2020 due to the lockdowns at the beginning of the pandemic and the first surge of cases.

At enrollment, the average age was 73, 49% female, 87% Black, average KCCQ of 62 (SD 24) and six-minute walk distance (6MWD) of 269 meters (+/- 118m), and NYHA I, II, III, IV distribution of 26%, 51%, 22%, 2%. Most baseline characteristics were similar between the cases and controls (Table 1). The ADI did not significantly differ between the groups. The cases were more likely to utilize Medicaid insurance (64% vs 39%, p = 0.002). They were less likely to identify as black (78% vs 90%, p = 0.034) or smoke (0% vs 15%, p = 0.004). For medications, the cases were more likely to take an ACE inhibitor (78% vs 62%, p 0.04), but there was no significant difference in loop diuretics, thiazides, beta-blockers, aldosterone antagonists, and digoxin use. COVID-19 infection was not associated with NYHA class, the short physical performance battery test or 6MWD, the overall KCCQ survey, nor any findings on the physical exam, ECG, or echocardiogram. For outcomes, average follow up time was 9.6 months. Deaths (6% vs 2.3%, p = 0.216) and number of hospitalizations (0.54 vs 0.37, p = 0.790) were not different

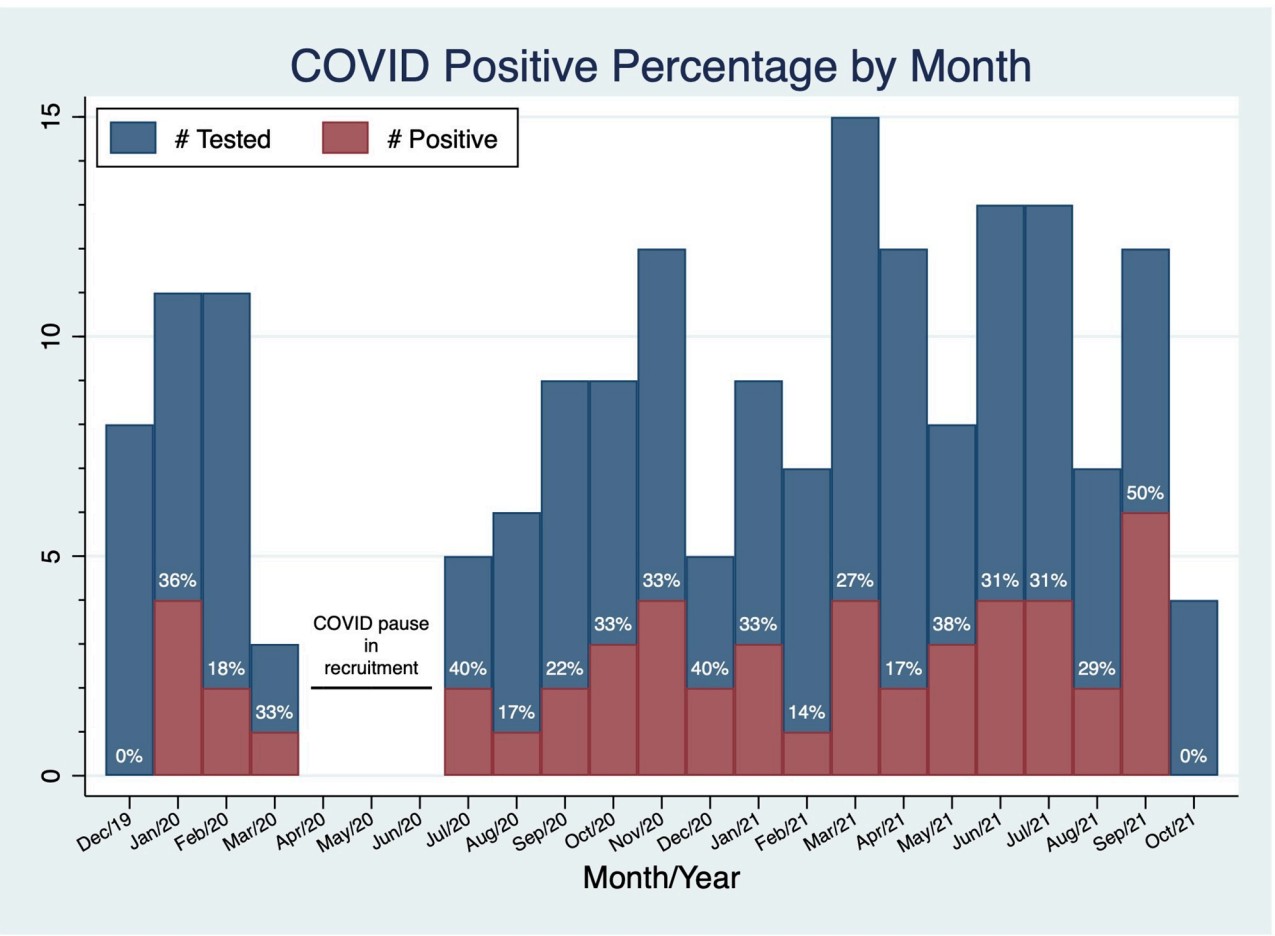

**Fig 2. Histogram of SARS-CoV-2 infections by month of enrollment (both NYC and Boston sites).** Cases are illustrated as red shaded area and total number of tested samples/enrolled subjects shaded in blue. The SCAN-MP study did not enroll during a 3 month period corresponding to the first COVID-19 surge. Note the high proportion of positive cases in January 2020, prior to recognition of community spread.

between cases and controls. None of the deaths were directly attributed to COVID-19 (Table 2). There were two unknown causes of death, 2 cases of complicated pneumonia, one case of pulmonary edema, and one respiratory arrest. The unadjusted and adjusted-for-age hazard ratios for death from SARS-CoV-2 infection were 2.94 (CI: 0.59–14.63, p = 0.198) and 3.03 (CI: 0.61–15.12, p = 0.276), respectively.

Among the 162 subjects who completed the questionnaire, there were no significant differences between cases and controls. Both cases and controls had similar attitudes towards COVID-19 risks of infecting their selves and others, behavior modifications, living conditions, outlook, and had similar sources for COVID-19 information (Fig 3A–3D and S1–S3 Figs). However, positive cases did report higher use of hand sanitizer (95.1% vs 82.5%, p 0.046).

Our study sought to (1) define the prevalence of COVID-19 infection among older, urban-dwelling Black and Hispanic patients during the initial phases of the pandemic in two of the most impacted locales—New York City and Boston, (2) to determine whether medical and social factors were associated with infection, and (3) assess whether attitudes or living conditions were associated with the likelihood of infection. Our findings demonstrate that COVID-19 was circulating in the NYC community prior to the first recognized case (before Jan 17, 2020) and that 44% (CI:14%-79%, binomial exact) of our cohort at our NYC sites were infected

**Table 1. Baseline characteristics, functional status, and outcomes.**

| | All (n = 180) | Positive* (n = 50) | Negative (n = 130) | p-value |
|---|---|---|---|---|
| **Baseline characteristics** | | | | |
| Age | 73 (9.2) | 73.4 (8.8) | 72.7 (9.4) | 0.659 |
| Male | 92 (51%) | 24 (48%) | 68 (52%) | 0.605 |
| Black | 156 (87%) | 39 (78%) | 117 (90%) | 0.034 |
| Hispanic | 58 (32%) | 20 (40%) | 38 (29%) | 0.166 |
| Site—NYC | 108 (60%) | 40 (80%) | 68 (52%) | 0.001 |
| Smoker (current) | 20 (11%) | 0 (0%) | 20 (15%) | 0.004 |
| Alcohol use | 56 (31%) | 12 (24%) | 44 (34%) | 0.229 |
| BMI | 31.4 (9.1) | 31.1 (9.7) | 31.4 (8.3) | 0.773◊ |
| **Medications** | | | | |
| Loop Diuretic | 108 (61%) | 27 (54%) | 81 (64%) | 0.206 |
| Beta Blocker | 90 (51%) | 28 (56%) | 62 (49%) | 0.832 |
| ACE-I/ARB | 117 (66%) | 39 (78%) | 78 (62%) | 0.040 |
| MRA | 24 (14%) | 7 (14%) | 17 (14%) | 0.929 |
| Digoxin | 7 (4%) | 3 (6%) | 4 (3%) | 0.387 |
| **NYHA class** | | | | 0.714 |
| I | 46 (26%) | 14 (28%) | 32 (25%) | |
| II | 91 (51%) | 27 (54%) | 64 (49%) | |
| III | 39 (22%) | 8 (16%) | 31 (24%) | |
| IV | 4 (2%) | 1 (2%) | 3 (2%) | |
| **Comorbidities** | | | | |
| Hypertension | 170 (94%) | 47 (94%) | 123 (95%) | 0.872 |
| DM | 113 (63%) | 27 (54%) | 86 (66%) | 0.131 |
| CKD | 53 (29%) | 13 (26%) | 40 (31%) | 0.529 |
| COPD | 31 (19%) | 7 (14%) | 27 (21%) | 0.299 |
| Total number | 5.6 (2.1) | 5.3 (1.9) | 5.7 (2.1) | 0.265 |
| **Social determinants of health** | | | | |
| Medicaid Insurance | 82 (46%) | 32 (64%) | 50 (38%) | 0.002 |
| ADI | 4 (4) | 4 (3) | 5 (4) | 0.209◊ |
| Health literacy score | 2 (1) | 2 (1) | 2 (1) | 0.889◊ |
| Trust in healthcare providers | 69 (8) | 69 (10) | 67 (6) | 0.063◊ |
| Perceived discrimination | 6 (5) | 6 (5) | 7 (6) | 0.283◊ |
| **Functional status** | | | | |
| SPPB | 8 (4) | 9 (4) | 8 (4) | 0.140◊ |
| 6MWT (meters) | 269 (118) | 256 (123) | 274 (116) | 0.414 |
| KCCQ Overall | 62.5 (40.6) | 63 (41.1) | 62.5 (40.3) | 0.762◊ |
| **Outcomes** | | | | |
| Hospitalized, ≥ 1 time | 84 (47%) | 24 (48%) | 60 (46%) | 0.824 |
| Hospitalized, ≥ 2 times | 34 (19%) | 11 (22%) | 23 (18%) | 0.508 |
| Deaths | 6 (3.3%) | 3 (6%) | 3 (2.3%) | 0.216 |

Reported as mean (sd) or median (IQR) for continuous; n(%) for categorical

* 40 (80%) by antibody test, 10 (20%) by PCR test

** 6 months before baseline visit and/or during follow up

◊ Mann-Whitney test due to lack of normal distribution

Abbreviations: NYC, New York City; BMI, body mass index; ACE-I, angiotensin-converting enzyme inhibitor; ARB, angiotensin II receptor blockers; MRA, mineralocorticoid receptor antagonists; NYHA, New York Heart Association; DM, diabetes mellitus; CKD, chronic kidney disease; COPD, chronic obstructive pulmonary disease; ADI, area deprivation index; SPPB, short physical performance battery; 6MWT, six-minute walk test; KCCQ, Kansas City cardiomyopathy questionnaire

**Table 2. Causes of death.**

| | Cause of Death |
|---|---|
| COVID Positive | 1. Unknown reason for death |
| | 2. Complicated pneumonia |
| | 3. Pulmonary edema |
| COVID Negative | 1. Unknown reason for death |
| | 2. Respiratory arrest |
| | 3. Complicated pneumonia |

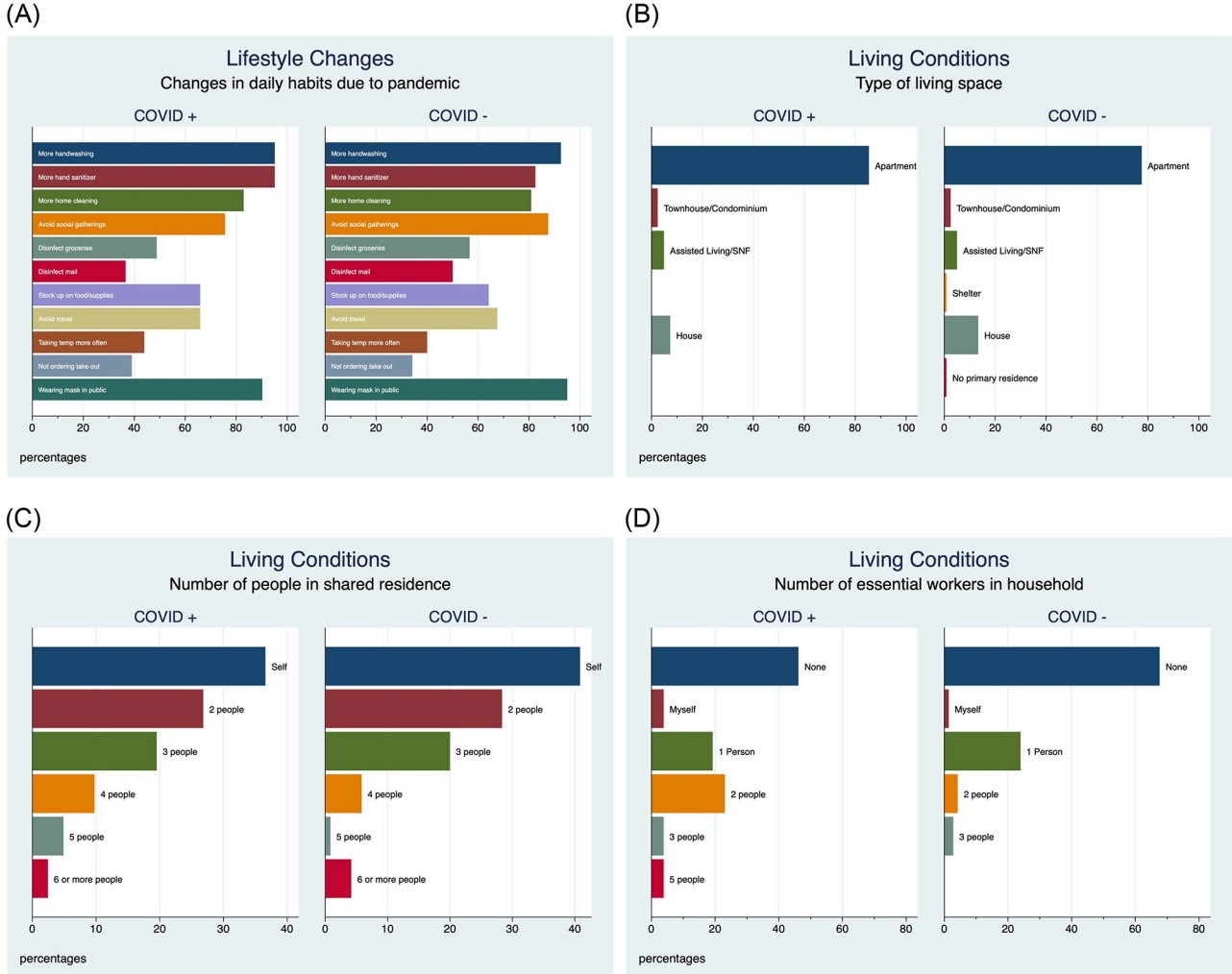

**Fig 3.** A. Lifestyle changes. B. Living conditions, type of living space. C. Number of people in shared living space. D. Number of essential workers in household. Participant responses to a generated questionnaire that assessed living conditions and lifestyle changes following the COVID-19 pandemic is illustrated. No significant differences were observed between cases and controls in attitudes following the onset of the pandemic nor baseline living conditions.

with SARS-CoV-2 in January 2020. We did not observe differences in medical co-morbidities but did identify sociodemographic factors may be associated with infection.

Prior studies that evaluated COVID-19 morbidity and mortality in heart failure patients have demonstrated that this subset of the population has not fared well when hospitalized [3–5, 10, 21]. From these retrospective and, almost-exclusively, hospital-based studies it is difficult to know what sort of risks heart failure patients in the community face in respect to infection. After SARS-CoV-2 infection, many remain asymptomatic, others experience mild symptoms, while the remainder experience more serious symptoms requiring hospitalization. Our study provides better insight into the former two categories of SARS-CoV-2 infection. In interpreting the results of our study, it is important to remember that 80% of our cases were found to have SARS-CoV-2 exposure based upon positive serum antibodies without recognized symptomatic infection. It can be assumed that many had been asymptomatic (although, it is possible that the nucleocapsid antibody may not detected in those with asymptomatic SARS-CoV-2 infection) [22], but it is also important to remember that many were positive before diagnostic tests were widely available and even before the virus was known to be in the United States, and certainly in NYC. The remaining 20% of our cases, all of whom had negative baseline serum antibody tests, were symptomatic and subsequently tested positive by PCR.

We can see from the nucleocapsid antibody tests that SARS-CoV-2 was widely circulating through New York City well before the first cases of COVID were documented in the US [23, 24]. Serologic testing from US Blood Donations has identified positive cases as early December 13-16th in West Coast states and early January in East Coast states [25]. From our study, it is remarkable to see that almost half (44%) of our NYC cohort that tested positive had been exposed to SARS-CoV-2 by the end of January, 2020. While this sample reflects a small subset of NYC's population, drawn from patients served by Columbia University Irving Medical Center and Harlem Hospital, it is likely that there was community spread of the virus to some significant degree which propagated infection. In the subsequent months (Fig 2), it is difficult to determine when the patients were infected with SARS-CoV-2, as it could have been at any time before their positive antibody titer. Also, of all the patients with positive antibodies, none of them later presented with symptoms for PCR testing for the duration of our study. This suggests that they maintained a strong adaptive immune response to SARS-CoV-2 after their first infection, through October of 2021.

The outcomes of death and hospitalizations did not significantly differ between cases and controls, yet the study was underpowered to detect a difference based on published mortality. Eighty percent of our cases were non-incident cases of prior SARS-CoV-2 infection, and this needs to be considered when interpreting the outcomes. Those who were hospitalized and died from COVID would not have subsequently enrolled in our heart failure cohort.

One finding that we found difficult to reconcile was the strong negative association of active smoking status with SARS-CoV-2 infection. This finding has been seen in prior studies and has come to be known as the "smokers paradox" [26–31]. A meta-analysis of 13 studies (n = 5960 patients) demonstrated that the pooled prevalence of smokers among hospitalized patients with COVID-19 in China was 6.5%, far lower than the prevalence of smoking in the overall Chinese population(26.6%) [28]. Similarly, a study from France demonstrated that current smokers were significantly underrepresented among COVID-19 patients, with only 4.1% of inpatients and 6.1% of outpatients, again far below the age and gender adjusted proportion of smokers in the general French population (24%) [31]. Our study differs from these prior reports in that our patients drawn from the same overall cohort, and thus infected cases and controls are demographically and medically similar, as may not occur when comparing hospitalized patients to the general population. Additionally, we show that smoking is negatively

associated with SARS-CoV-2 infection, and not symptomatic COVID-19 infection, as is the case with these prior studies.

Several biological mechanisms by which smoking might be protective in COVID-19 have been proposed, including increased ciliary beat frequency within the ciliated epithelium of the nasal tract, anti-inflammatory effect of nicotine, and increased nitric oxide in the respiratory tract with may inhibit the replication of SARS-CoV-2 and entry into cells [26, 32–35]. Another proposed mechanism is that smoking may upregulate ACE2, which may counteract the viral degradation of the enzyme (which it uses for cell-entry), and thereby protect against angiotensin II overactivity (by converting it to angiotensin 1–7) after SARS-CoV-2 infection. Behavioral adaptations to smoking, such as seeking ventilation or outdoor space to smoke may also have contributed to lower infection rates.

While less overwhelming than the differences in smokers, the use of ACE-inhibitors differed between cases and controls by 16% (absolute percentage difference). At the outset of the pandemic, there was a question of COVID risk with ACE-inhibitor usage, since there is upregulation of the ACE2 receptor in patients taking ACE-inhibitors and ARBs, thereby potentially facilitating virus enter into cells [36, 37]. While multiple studies have shown that ACE-inhibitors/ARBs are not associated with COVID morbidity and mortality, there is not much evidence for or against an association with SARS-CoV-2 infection as it has only been evaluated in cases of symptomatic COVID [38–41]. It may be that there is a slight increased risk of SARS-CoV-2 infection in patients taking ACE-inhibitors/ARBs, but additionally they may be protected from developing symptomatic or even severe COVID.

ADI was not significantly different between cases and controls. We had expected that our SARS-CoV-2 infected patients would live in districts with lower average socio-economic status. In surveying the literature, however, several studies have found that ADI increases odds of COVID infection but this is seen in rural areas, not urban [42–44]. The reason for this is not clear, but it is possible that ADI, which is often used as a surrogate for individual level socio-economic status may lose individual-level socio-economic resolution in higher density population census blocks. Smaller numbers of individuals within an ADI block may make it less susceptible to ecological fallacy, which is the assumption that inferences can be made about individual patterns based on patterns observed in groups [45, 46]. The fact that Medicaid utilization was significantly higher in among those infected with SARS-CoV-2 would support the idea that ADI may not be an effective surrogate for socioeconomic status in urban areas. Another possibility is that the racial makeup of rural America is very different than urban areas, where there may be different attitudes toward social distancing and other guidelines from the pandemic. Additionally, the association of ADI with COVID-19 may be temporal based on decreased vigilance, as a study in Belgium found that assessed socioeconomic deprivation increased the odds of COVID-19 infection but only in their second and third waves, but not the first wave [44].

As part of our study, we wanted to describe differences in behaviors, attitudes, and living situations between those who were infected and those who were not. The survey we administered was comprehensive but there were almost no differences between the cases and controls.

Our study has several important limitations. From an internal validity standpoint, our numbers were too small to detect a difference in hospitalizations and death. Additionally, patients were only questioned about hospitalizations 6 months prior to the start of the study, but for anyone enrolled at or after July of 2020 (6 months after the start of the pandemic), COVID related hospitalizations could have been missed. Additionally, survival bias attenuates the effect size of hospitalization and death. From this standpoint, associations between our cases and outcomes can best be interpreted on how recent (not incident) SARS-CoV-2 infection is associated with death and hospitalization. From a general validity perspective, our

cohort is restricted to self-identified Black or Hispanic patients in an urban setting, and some of our findings may not be applicable to populations comprised of different racial/ethnic and socioeconomic background. Finally, our questionnaire assessing attitudes was administered while the virus was circulating and exposure could have occurred, thus the timing of infection may have occurred prior to the questionnaire administration.

The strengths of our study are that it is largely prospective (even though a portion were identified retrospectively) and that using nucleocapsid antibodies to SARS-CoV-2 we could identify associations with prior infection that may not have been detected if our cases were only those with COVID-19 symptoms. Indeed many COVID-19 cases went undetected during the first wave due to lack of testing availability and using serology our study demonstrates that the prevalence of community spread was higher and earlier than conventionally known.

## Conclusions

Among a community cohort of older self-identified Black or Hispanic patients with heart failure living in New York City and Boston, MA, USA, we observed that 28% had evidence of SARS-CoV-2 infection. The first case in our study was identified prior to the first reported case in New York City, suggesting that the virus was circulating in early January 2020. There was no increased mortality or hospitalizations among those infected with SARS-CoV-2.

## Supporting information

**S1 Fig. Perceived risk.** KEY: (a)"You will be infected" (b)"Someone in your direct environment (family, friends, or colleagues) will be infected" (c)"You will have to go to the hospital if you get infected" (d)"You will have to go into quarantine" (e)"You will get infected and you will infect someone else" (f)"Someone in your direct circle of people (family, friends, colleagues) will become ill or die". The hollow circles represent subject selection of answers. Red circle represents the average(mean) response, with the horizontal line representing the 95% confidence interval.
(TIF)

**S2 Fig. Perceived effectiveness of interventions.** KEY: (a)"Wearing a mask" (b)"Washing your hands with soap or using hand sanitizer frequently" (c)"Seeing a health care provider if you feel sick" (d) "Seeing a health care provider if you feel healthy but worry that you were exposed" (e)"Avoiding public spaces, gatherings, and crowds" (f) "Avoiding contact with people who could be high-risk" (g) "Avoiding hospitals and clinics" (h) "Avoiding restaurants" (i)"Avoiding public transit". The hollow circles represent subject selection of answers. Red circle represents the average(mean) response, with the horizontal line representing the 95% confidence interval.
(TIF)

**S3 Fig. Source of COVID information.**
(TIF)

**S1 File. COVID-19 questionnaire.**
(DOCX)

## Author Contributions

**Conceptualization:** Elizabeth G. Cohn, Sergio L. Teruya, Stephen Helmke, Denise Fine, Codruta Chiuzan, Mathew S. Maurer, Frederick L. Ruberg.

**Data curation:** Sergio L. Teruya, Stephen Helmke, Denise Fine, Frederick L. Ruberg.

**Formal analysis:** Jonathan B. Edmiston, Frederick L. Ruberg.

**Funding acquisition:** Denise Fine, Mathew S. Maurer, Frederick L. Ruberg.

**Investigation:** Elizabeth G. Cohn, Varsha Muralidhar, Codruta Chiuzan, Mathew S. Maurer.

**Methodology:** Jonathan B. Edmiston, Elizabeth G. Cohn, Sergio L. Teruya, Codruta Chiuzan, Eldad A. Hod, Mathew S. Maurer, Frederick L. Ruberg.

**Project administration:** Sergio L. Teruya, Natalia Sabogal, Daniel Massillon, Carlos Rodriguez, Denise Fine, Morgan Winburn, Farbod Raiszadeh, Mathew S. Maurer, Frederick L. Ruberg.

**Resources:** Denise Fine, Eldad A. Hod, Frederick L. Ruberg.

**Software:** Jonathan B. Edmiston, Sergio L. Teruya, Denise Fine.

**Supervision:** Natalia Sabogal, Stephen Helmke, Morgan Winburn, Farbod Raiszadeh, Damien Kurian, Mathew S. Maurer, Frederick L. Ruberg.

**Validation:** Sergio L. Teruya, Eldad A. Hod.

**Visualization:** Jonathan B. Edmiston.

**Writing – original draft:** Jonathan B. Edmiston, Frederick L. Ruberg.

**Writing – review & editing:** Jonathan B. Edmiston, Elizabeth G. Cohn, Sergio L. Teruya, Varsha Muralidhar, Stephen Helmke, Denise Fine, Codruta Chiuzan, Mathew S. Maurer, Frederick L. Ruberg.

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
