## [Decision Letter · Decision Letter 0]

22 Nov 2022

PONE-D-22-19779Clinical and social determinants of health features of SARS-CoV-2 infection among Black and Caribbean Hispanic patients with heart failure: The SCAN-MP StudyPLOS ONE

Dear Dr. Ruberg,

Thank you for submitting your manuscript to PLOS ONE. After careful consideration, we feel that it has merit but does not fully meet PLOS ONE’s publication criteria as it currently stands. Therefore, we invite you to submit a revised version of the manuscript that addresses the points raised during the review process.

We look forward to receiving your revised manuscript.

Kind regards,

Dong Keon Yon, MD, FACAAI

Academic Editor

PLOS ONE

Journal Requirements:

3. Please expand the acronym “NHLBI” (as indicated in your financial disclosure) so that it states the name of your funders in full.

"This study was funded by NIH/NHLBI R01 HL139671 and HL139671S1 (to FLR and MSM)."

"This study was funded by NIH/NHLBI R01Hl139671 and R01HL139671S1 to FLR and MSM. "

6. Please clarify the Tables 1 and 2 in your manuscript and separate supporting information tables file.

Additional Editor Comments:

Thank you for submitting your manuscript. The reviewers and I believe it is of potential value for our readers. However, the reviewers have raised a number of very important issues, and their excellent comments will need to be adequately addressed in a revision before the acceptability of your manuscript for publication in the Journal can be determined. We cannot guarantee that your revised paper will be chosen for publication; this would be solely based on how satisfactorily you have addressed the reviewer comments.

Reviewers' comments:

Reviewer's Responses to Questions

**Comments to the Author**

1. Is the manuscript technically sound, and do the data support the conclusions?

Reviewer #1: Yes

Reviewer #2: Partly

Reviewer #3: Yes

2. Has the statistical analysis been performed appropriately and rigorously? 

Reviewer #1: Yes

Reviewer #2: Yes

Reviewer #3: Yes

3. Have the authors made all data underlying the findings in their manuscript fully available?

Reviewer #1: Yes

Reviewer #2: Yes

Reviewer #3: Yes

4. Is the manuscript presented in an intelligible fashion and written in standard English?

Reviewer #1: Yes

Reviewer #2: Yes

Reviewer #3: Yes

5. Review Comments to the Author

Reviewer #1: The (Screening for Cardiac Amyloidosis with Nuclear Imaging )SCAN-MP study aims to assess the overall prevalence of transthyretin cardiac amyloidosis in African Americans and Hispanic Caribbean with heart failure. That is, this study excluded heart failure (HF) patients due to left-sided valvular disease or ischemic cardiomyopathy. This manuscript sought to characterize the medical and non-medical factors associated with SARS-CoV-2 infection in this specific HF group.

Their hypothesis was that infection rates would be the highest in this entire minority HF population. This study contained two cohorts, one retrospective cohort (55 patients) and another prospective cohort (125 patients). There were two main findings: 1. COVID-19 was circulating in the NYC community prior to the first recognized case. 2. SARS-CoV-2 infection was common in this minority HF population. Another two findings: 1. The rate of mortality or hospitalization was not increased among patients infected with SARS-CoV-2. 2. Health literacy and ADI were also not associated with infection. The most important limitation regarding these conclusions is too small study numbers.

Personally, I agree with this manuscript being published in its current state.

Reviewer #2: The present study investigates a cohort of SARS-CoV-2 infection patients with the main focus on whether medical, social factors, attitudes, and living conditions were associated with infection, and found that SARS-CoV-2 infection was common among older, minority patients with HF and not associated with Health literacy and ADI. Overall, the work here is innovative, and the article is written well.

However, the Screening for Cardiac Amyloidosis Using Nuclear Imaging for Minority Populations (SCAN-MP) study is an NHLBI-funded, prospective cohort study that is enrolling Black or Caribbean Hispanic participants over the age of 60 years with heart failure. Therefore, the conclusions of the study are limited to particular patients. For instance, as shown in your results, deaths (6% vs. 2.3%, p=0.216) and the number of hospitalizations (0.54 vs. 0.37, p=0.790) were not different between infected and non-infected groups. The impact of competing events (primary disease: heart failure) might be more significant than COVID-19 for prognosis. Besides, the sample number was relatively small, which to a certain extent, reduces the credibility of your results. I, therefore, suggest that addressing the following would strengthen the manuscript:

1. Clinically used classic myocardial injury indicators, which can effectively reflect the myocardial injury of patients, could be added to the analysis of the outcome of SARS-CoV-2 infection patients with heart failure.

2. NYHA class changes before and after infection should be described to demonstrate no increased mortality or hospitalizations among those infected with SARS-CoV-2.

3. Another limitation is the absence of the number of severe and non-severe COVID-19, which may have affected outcomes.

Reviewer #3: Dear authors,

I have now completed the review of the manuscript titled "Clinical and social determinants of health features of SARS-CoV-2 infection among Black and Caribbean Hispanic patients with heart failure: The SCAN-MP Study."

In the present study, the authors characterized medical and non-medical factors associated with SARS-CoV-2 infection.

The manuscript is interesting and, in general, fair written.

I have one minor suggestion before recommending acceptance.

In the ‘Statistical Analysis’ section, some sentences needs reference as follows:

Means for continuous variables were compared using independent group t-tests when data were normally distributed; otherwise, Mann-Whitney tests were performed[1].

The hazard ratio for death was calculated using a Cox proportional hazards model with the time of follow up starting from baseline antibody test[2] ...

[1] https://doi.org/10.54724/lc.2022.e1

[2] https://doi.org/10.54724/lc.2022.e4

I would be glad if you could do this and resend me a revised manuscript.

Thank you.

6. PLOS authors have the option to publish the peer review history of their article (what does this mean?). If published, this will include your full peer review and any attached files.

Reviewer #1: **Yes: **Shyh-Ming Chen

Reviewer #2: No

Reviewer #3: No

---

## [Author Response · Author response to Decision Letter 0]

4 Jan 2023

We thank the reviewers for their valuable feedback and have modified our manuscript accordingly. We have also edited the manuscript to conform to Journal style as requested.

Reviewer #2:

1) Clinically used classic myocardial injury indicators, which can effectively reflect the myocardial injury of patents, could be added to the analysis of the outcome of SARS-CoV-2 infection patients with heart failure.

We agree with the reviewer’s point, but unfortunately, we do not have myocardial injury markers available at this time. Injury markers, including high sensitivity troponin I and T, will be measured at the completion of the entire SCAN-MP study in 2024 or 2025 in batch through a contract with Abbott Laboratories. We might add that injury markers are likely to be abnormal given that these patients have chronic heart failure that commonly is associated with low level elevations in high sensitivity markers.

2) NYHA class changes before and after infection should be described to demonstrate no increased mortality or hospitalizations among those infected with SARS-CoV-2.

We agree that having pre and post infection NYHA class could be useful, but unfortunately the design of our study is a baseline formal assessment of NYHA class (as we have presented) with limited phone followup. We do not have information regarding serial change in NYHA class.

3) Another limitation is the absence of the number of the severe and non-severe COVID-19, which may have affected outcomes. 

We agree that this is an important limitation of our study and have clarified this in the limitation section of our manuscript with the following:

“Additionally, patients were only questioned about hospitalizations 6 months prior to the start of the study, but for anyone enrolled at or after July of 2020 (6 months after the start of the pandemic), COVID related hospitalizations could have been missed. Additionally, survival bias attenuates the effect size of hospitalization and death.” 

Thank you for the opportunity to respond to these thoughtful comments.

Reviewer #3

References have been added as requested – new references 19 and 20.

---

## [Decision Letter · Decision Letter 1]

15 Mar 2023

Clinical and social determinants of health features of SARS-CoV-2 infection among Black and Caribbean Hispanic patients with heart failure: The SCAN-MP Study

PONE-D-22-19779R1

Dear Dr. Ruberg,

We’re pleased to inform you that your manuscript has been judged scientifically suitable for publication and will be formally accepted for publication once it meets all outstanding technical requirements.

Kind regards,

Dong Keon Yon, MD, FACAAI

Academic Editor

PLOS ONE

Additional Editor Comments (optional):

This is an excellent paper.

Reviewers' comments:

Reviewer's Responses to Questions

**Comments to the Author**

1. If the authors have adequately addressed your comments raised in a previous round of review and you feel that this manuscript is now acceptable for publication, you may indicate that here to bypass the “Comments to the Author” section, enter your conflict of interest statement in the “Confidential to Editor” section, and submit your "Accept" recommendation.

Reviewer #2: All comments have been addressed

Reviewer #3: All comments have been addressed

2. Is the manuscript technically sound, and do the data support the conclusions?

Reviewer #2: Yes

Reviewer #3: Yes

3. Has the statistical analysis been performed appropriately and rigorously? 

Reviewer #2: Yes

Reviewer #3: Yes

4. Have the authors made all data underlying the findings in their manuscript fully available?

Reviewer #2: Yes

Reviewer #3: Yes

5. Is the manuscript presented in an intelligible fashion and written in standard English?

Reviewer #2: Yes

Reviewer #3: Yes

6. Review Comments to the Author

Reviewer #2: (No Response)

Reviewer #3: All comments have been addressed. Thank you to the authors and editors for considering my opinion on this manuscript.

7. PLOS authors have the option to publish the peer review history of their article (what does this mean?). If published, this will include your full peer review and any attached files.

Reviewer #2: No

Reviewer #3: No

---

## [Editor Report · Acceptance letter]

21 Mar 2023

PONE-D-22-19779R1 

Clinical and social determinants of health features of SARS-CoV-2 infection among Black and Caribbean Hispanic patients with heart failure: The SCAN-MP Study 

Dear Dr. Ruberg:

I'm pleased to inform you that your manuscript has been deemed suitable for publication in PLOS ONE. Congratulations! Your manuscript is now with our production department. 

Kind regards, 

on behalf of

Dr. Dong Keon Yon 

Academic Editor

PLOS ONE